# A-REST (Activity to Reduce Excessive Sitting Time): A Feasibility Trial to Reduce Prolonged Sitting in Police Staff

**DOI:** 10.3390/ijerph19159186

**Published:** 2022-07-27

**Authors:** Marsha L. Brierley, Lindsey R. Smith, Angel M. Chater, Daniel P. Bailey

**Affiliations:** 1Institute for Sport and Physical Activity Research, School of Sport Science and Physical Activity, University of Bedfordshire, Polhill Avenue, Bedford MK41 9EA, UK; marsha.brierley@gmail.com (M.L.B.); lindsey.smith@beds.ac.uk (L.R.S.); a.chater@ucl.ac.uk (A.M.C.); 2Division of Sport, Health and Exercise Sciences, Department of Life Sciences, Brunel University London, Uxbridge UB8 3PH, UK; 3Centre for Physical Activity in Health and Disease, Brunel University London, Kingston Lane, Uxbridge UB8 3PH, UK; 4Centre for Behaviour Change, University College London, 1-19 Torrington Place, London WC1E 7HB, UK

**Keywords:** sitting, intervention, feasibility, office workers, behaviour change wheel, police, QR codes, activity breaks

## Abstract

The aim of this study was to evaluate the acceptability and feasibility of a theory-derived sedentary workplace intervention for police office staff. Twenty-four staff participated in an 8-week intervention (single arm, pre-post design) incorporating an education session, team competition with quick response (QR) codes, team trophy, weekly leaderboard newsletters, a self-monitoring phone app, and electronic prompt tools. The intervention supported participants to reduce and break up their sitting time with three minutes of incidental movement every 30 min at work. Feasibility and acceptability were assessed using mixed methods via the RE-AIM QuEST and PRECIS-2 frameworks. The intervention was highly pragmatic in terms of eligibility, organisation, adherence, outcome, and analysis. It was slightly less pragmatic on recruitment and setting. Delivery and follow-up were more explanatory. Reach and adoption indicators demonstrated feasibility among police staff, across a range of departments, who were demographically similar to participants in previous office-based multi-component interventions. The intervention was delivered mostly as planned with minor deviations from protocol (implementation fidelity). Participants perceived the intervention components as highly acceptable. Results showed improvements in workplace sitting and standing, as well as small improvements in weight and positive affect. Evaluation of the intervention in a fully powered randomised controlled trial to assess behaviour and health outcomes is recommended.

## 1. Introduction

Office workers are sedentary for approximately 77–82% of an 8 h workday and 70–76% of their overall day [1,2]. Chronic disease risk and all-cause mortality increases with high levels of sitting time [3,4,5,6]. As society moves further towards increased use of technology and computer work [7], this poses a significant risk to occupational and public health. Interventions to reduce workplace sitting have been effective in reducing sitting by 30 to 120 min per workday [8,9] and show promise for improving cardiometabolic health [10]. It is also suggested that sedentary workplace interventions may produce the best results for individuals who are most sedentary and at risk of chronic disease [11]. In a large cross-sectional investigation into the occupational characteristics of British police force employees (*n* = 5527), at least 30% were found to hold traditionally sedentary work roles (i.e., desk-based office work) [12]. Police staff and those of lower rank (constable/sergeant) are more likely to have adverse cardiometabolic health profiles than those in higher ranks [13]. Thus, this represents an occupational group that could benefit from an intervention to reduce sitting. Sedentary workplace interventions have previously targeted individual [14,15,16], environmental [17,18,19,20], and/or a combination of individual, environmental, and organisational-level components [21,22,23,24]. Multi-component interventions that incorporate individual, organisational, and environmental changes, and environmental interventions that use active workstations, are the most effective for reducing workplace sitting [9,25,26]. At present, the evidence quality for environmental-only interventions remains weak and it is unclear how well sitting reductions are sustained over the long term [25,26]. Multi-level approaches are likely to demonstrate greater behaviour effects than single level interventions (e.g., individual-level or organisation-level only) [27] and are consistent with best practice for complex interventions [28,29]. A systematic review of twenty-one health promotion studies in the police (targeting physical activity, diet, lifestyle, sedentary behaviour or a combination of these) found that multi-component interventions with education and behaviour change components, and interventions that involved peer support, had the greatest impact on health outcomes [30]. Further investigation into the efficacy of multi-component sedentary workplace interventions with police staff is warranted.

To better understand how interventions work and why, improved evaluation methods are required, particularly around external validity [31]. Two frameworks to aid in this evaluation are PRECIS-2 (PRagmatic Explanatory Continuum Indicator Summary) [32,33] and RE-AIM QuEST (RE-AIM: Reach, Effectiveness, Adoption, Implementation, Maintenance, QuEST: Qualitative Evaluation for Systematic Translation) [34,35]. PRECIS-2 is specifically intended for the feasibility/design stage and helps researchers understand trial effectiveness under usual (pragmatic) conditions. RE-AIM QuEST can be applied to all stages of study design and includes additional items on the generalisability and applicability of trials in specific contexts. Frameworks such as RE-AIM QuEST allow a better understanding of additional indicators of success or failure so they can be systematically defined, measured, and addressed for future implementation [36]. The use of both frameworks simultaneously can aid in the consistent reporting of key study characteristics in order to help researchers understand where a study is pragmatic, where is it explanatory, and how best to translate theory into practice [37]. This is particularly important for feasibility work intended to inform the design and execution of larger randomised controlled trials (RCTs).

The primary aim of this study, therefore, was to assess the acceptability and feasibility of an intervention to break up and reduce prolonged sitting in police staff. The secondary aims were to assess the potential effects of the intervention on workplace and total daily sitting, standing, and stepping, as well as cardiometabolic risk markers, psychological wellbeing, mood, work stress, job satisfaction, and work performance.

## 2. Materials and Methods

### 2.1. Study Design and Overview

This was a single arm, pre-post, repeated measures feasibility trial conducted in the UK between August 2019 and December 2019. Baseline assessments were carried out ≤3.5 weeks prior to intervention start. The intervention then ran for 8 weeks. In the final week of the intervention (week 8), device-measured sitting, standing and stepping were assessed. A quantitative (e.g., surveys, device-assessed behaviour, and body measurements) and qualitative (e.g., semi-structured interviews) mixed-methods design was used for data collection and evaluation. Cardiometabolic risk markers, psychological wellbeing, mood, work stress, job satisfaction, and work performance were assessed post-intervention (week 9). See the bottom of Figure 1. for the study timeline. The trial was prospectively registered at clinicaltrials.gov (NCT04053686). The study was approved by the University of Bedfordshire Institute for Sport and Physical Activity Research Ethics Committee (2019ISPAR008) and was performed in accordance with the Declaration of Helsinki for research involving human participants. A CONsolidation Standards Of Reporting Trials (CONSORT) extension to randomised pilot and feasibility trials checklist [38] is provided in Appendix A. A detailed intervention protocol is provided using the Template for Intervention Description and Replication (TIDieR) [39] in Appendix A.

#### 2.1.1. Study Setting and Recruitment

The study took place at a single site, Bedfordshire police headquarters, in Kempston, UK. The site had a workforce of 2272 people of which 847 were police staff. Participant offices were located across the worksite, which was comprised of a large main building with three floors, a second smaller building, and a modular portacabin office. Satellite offices were not recruited for the study.

Participant recruitment took place August 2019 to October 2019. Suitable departments comprising ~175–200 employees were initially identified by management for invitation via email. The researchers also attended the worksite to set up study information booths to provide potential participants with information about the study and, later, study information posts on the workforce intranet were provided. Figure 1 shows the recruitment flowchart. Workers were encouraged to sign up with colleagues from the same office, but this was not a requirement. Interested individuals were guided through the enrolment process using an online system (Qualtrics Inc., Seattle, WA, USA) where they were provided with an information sheet, screened for eligibility, and gave informed consent.

#### 2.1.2. Sample Size

A sample size in the range of 24–50 participants is suggested for feasibility trials [40,41,42]. Thus, a target of 30 participants was agreed by the research team as this was considered pragmatic and suitable for gathering sufficient data regarding trial feasibility and acceptability of the intervention while allowing for drop out.

#### 2.1.3. Eligibility Criteria

Eligible individuals met the following inclusion criteria: ≥18 years old; working ≥ 0.6 full-time equivalent hours [23]; ambulatory; predominantly desk-based (self-reported ≥ 5 h/day seated at work); able to keep a smartphone with them during work hours; and were apparently healthy. Individuals were excluded if they had a planned absence of two weeks or more during the intervention period; had health contraindications to standing or walking; had a planned relocation to another site, office or workplace during the study period; or had personal access to an active workstation.

#### 2.1.4. The A-REST Intervention

The A-REST (Activity to Reduce Excessive Sitting Time) intervention was developed using the Behaviour Change Wheel [43]. The eight-week intervention aimed to reduce and break up participants’ sitting with 3 min incidental movement breaks every half an hour at work. A-REST was comprised of an education session, a behaviour change booklet, electronic prompts, Quick Response (QR) codes to log breaks from sitting, a team competition with trophy, health champion, weekly emails, a smartphone app for self-monitoring and individual feedback on behaviour (see Figure 2). Behaviour change techniques (BCTs) from the BCT taxonomy (v1) [44] were selected using a three-stage process. First, promising BCTs for sitting reduction were identified in a systematic review of office workplace interventions that evaluated effects on cardiometabolic risk markers [10]. Second, police staff were interviewed to identify their capability, opportunity, and motivation for reducing sitting time at work [45]. Third, interviews on the acceptability and feasibility of potential strategies (such as sit-stand desks) with participants who had completed a separate workplace intervention using the Behaviour Change Wheel approach [46,47] guided the APEASE (Acceptability, Practicability, Effectiveness, Affordability, Side-effects, Equity) criteria to identify content and strategies in the current intervention. Twenty BCTs were included in the final intervention (see Appendix A).


Figure 1Flowchart detailing recruitment, response rates, baseline data collection attendance, and study timeline.
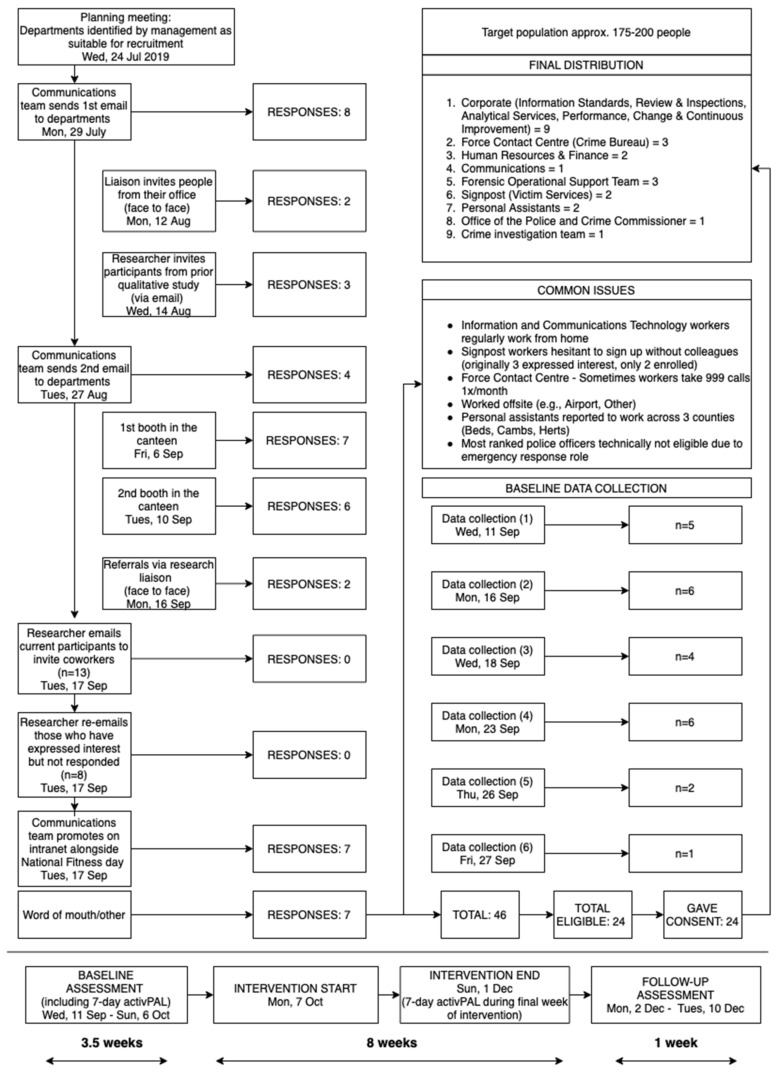




Figure 2A-REST (Activity to Reduce Excessive Sitting Time) intervention infographic [3,5,12,46].
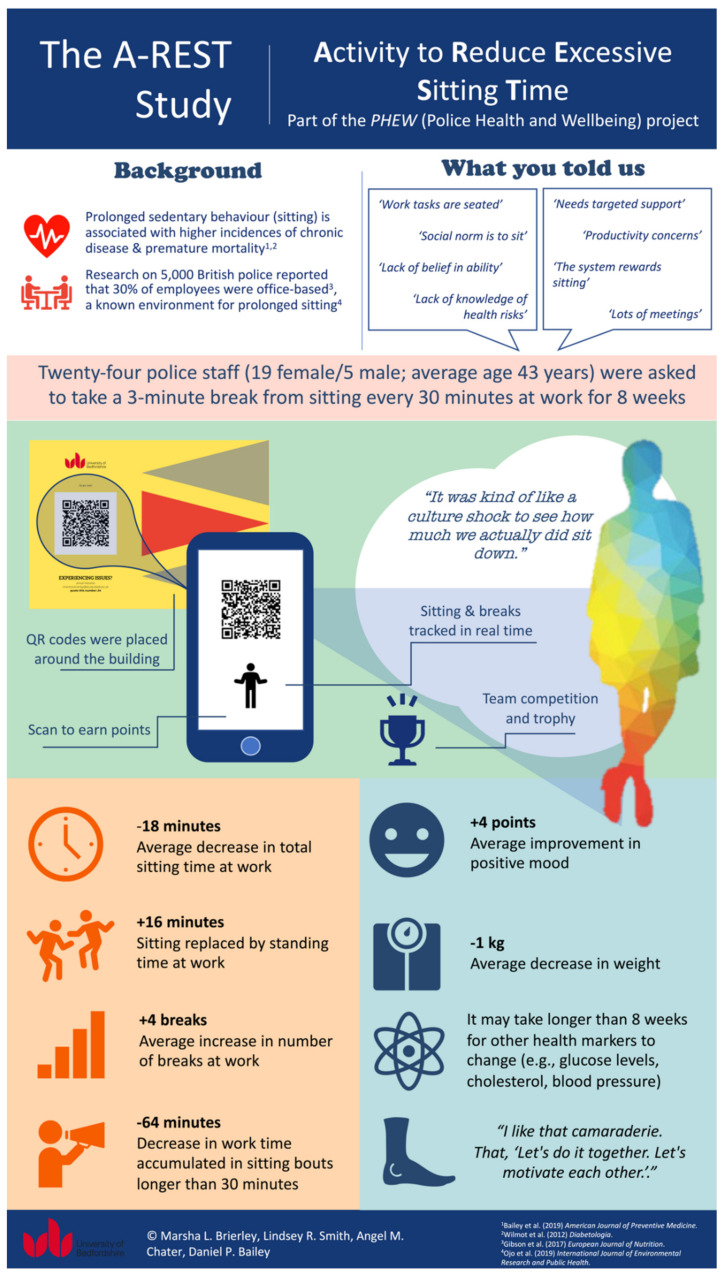



The A-REST intervention supported behaviour change with the following strategies. In the first week, participants attended a 45 min interactive education session delivered by the lead researcher (MLB). A-REST behaviour change booklets were provided and completed by participants at this time (see Appendix A). PowerPoint slides (Microsoft Corporation, Redmond, WA, USA) were presented that exactly corresponded to the pages in the A-REST booklet. Participants were asked to break up their sitting time every 30 min at work with 3 min incidental movement breaks.

A team competition was set up to help them achieve the set behavioural goal. Participants were put into teams and asked to use their smartphone to scan QR codes located around the worksite to log their individual breaks from sitting on workdays. Each scan earned a point for their team. A trophy reward went to the team with the most points each week. Participants were allocated to teams of three based on desk proximity. Three participants who were not in the same office as any other participants were grouped into a team together. Twenty-six QR codes were printed on laminated A5 paper and attached to walls around the site (see Appendix A). Participants were asked to log a break from sitting every half an hour with no additional instructions given. During the education session, participants had decided on placement of the QR codes, which resulted in assigning one to every team’s office as well as common areas such as the canteen, near toilets, and in stairwells. In addition, participants could request a personal QR code card that could fit on their identification lanyard (business card size) to log breaks from sitting during meetings when they felt unable to walk to other QR codes in the building. This was a suggestion from participants during the education session.

Participants received weekly email updates with competition leaderboard standings and tips. Emails were based on a template (see Appendix A) and personalised each week by either the research team or a health champion who was a Health and Wellbeing coordinator at the worksite and acted as research liaison for the study.

Participants were asked to download a smartphone app (Rise & Recharge^®^; Baker Heart & Diabetes Institute, Melbourne, Australia) so they could self-monitor their breaks from sitting. The app provided automatic tracking of breaks from sitting throughout the day. Number of daily breaks achieved were visually displayed in the app as well as weekly and monthly break achievements. This app also allows users to set up prompts to remind them to break up their sitting, but this feature was not functional for the duration of the study. A handout was provided to participants at the education session with a list of electronic prompts that they could select from (i.e., phone apps, computer prompts, electronic calendar and/or phone timers) to remind them to break up their sitting at regular intervals.

#### 2.1.5. Demographic Measurements

At baseline, demographic data, body mass index (BMI), self-reported sitting time [48], self-reported physical activity (International Physical Activity Questionnaire short form) [49], and prevalence of musculoskeletal complaints [50,51] were collected using a survey hosted on Qualtrics. The demographic questionnaire included items on age, gender, ethnicity, education, cohabitation status, job role, rank, years in service, hours worked per week, office size, medical conditions (hypertension, high cholesterol and/or high blood pressure), perceived health (self-rated 4-point scale of poor, good, very good, or excellent), smoking status [52] and alcohol intake [53]. The average metabolic equivalents of task (METs) were computed for walking, moderate and vigorous physical activity.

### 2.2. Outcome Measurements

#### 2.2.1. Primary Outcome—Feasibility and Acceptability

Two frameworks, PRECIS-2 and RE-AIM QuEST, were employed to evaluate trial feasibility and acceptability. Feasibility refers to whether an intervention can be implemented (as intended) within a particular context [54,55]. Acceptability has been defined as the degree to which an intervention is perceived as agreeable or satisfactory among stakeholders [54,55]. PRECIS-2 assesses how pragmatic an intervention is and helps intervention designers make consistent decisions in line with whether the study is designed to be pragmatic (delivered in a real-world setting) or explanatory (delivered under ideal conditions). PRECIS-2 was used to evaluate the study on nine dimensions (1. eligibility criteria, 2. recruitment, 3. setting, 4. organisation, 5. flexibility-delivery, 6. flexibility-adherence, 7. follow-up, 8. primary outcome, and 9. primary analysis). Each dimension was assessed on a 5-point Likert scale (1 = very explanatory to 5 = very pragmatic) [32]. This framework has acceptable internal reliability and validity [32,56].

The RE-AIM QuEST framework assesses five key indicators of successful implementation using mixed methods: Reach, Effectiveness, Adoption, Implementation, and Maintenance (see Table 1). Together, these domains describe the representativeness of the setting and participants, engagement with the intervention by setting/site, effectiveness, deviations from protocol, and sustainability. Qualitative post-intervention participant interview quotations were presented to provide specific answers around how and why the intervention worked (or did not work) to change behaviour. Acceptability was assessed under the Implementation indicator including participant retention, changes in behaviour (breaking up and reducing sitting), and thematic analysis of interviews [57]. Fidelity was planned *a priori* [58] and reported within the Implementation component of RE-AIM (see Table 1). RE-AIM has been used previously to assess the potential for scaling up of pilot sedentary behaviour interventions [59], and in the early adoption phase of BeUpstanding™ to identify improvements for wider implementation of the programme [60]. Assessing key RE-AIM indicators in a feasibility trial helps identify intervention elements requiring adjustment before implementation in a definitive RCT to evaluate its potential public health impact [61].

#### 2.2.2. Secondary Outcome—Potential Effects

This study also investigated potential effects on workplace and daily sitting, standing, and stepping, as well as cardiometabolic risk markers, psychological wellbeing, mood, work stress, job satisfaction, and work performance. These measures were taken at baseline and post-intervention (other than sitting, standing and stepping, which were measured during the final week of the intervention).

Sitting, standing, and stepping

An activPAL3 tri-axial accelerometer and inclinometer device (PAL Technologies, Glasgow, UK) was worn for seven consecutive days (24 h/day) to objectively measure sitting, standing and stepping. The activPAL was waterproofed and worn under the clothes on the midline of the anterior aspect of the thigh attached using a 10 cm × 10 cm adhesive dressing (Hypafix, BSN medical Limited, Yorkshire, UK). ActivPAL monitors provide a valid and reliable assessment of sitting, standing, postural transitions and stepping [62,63,64,65,66]. Participants recorded sleep/waking times, working hours, and non-wear time in a diary to aid with data processing [67]. Data were processed using Processing PAL v1.2 (University of Leicester, UK) and calibrated to waking and working hours from participants’ diaries. The algorithm used in the Processing PAL software has been validated among a representative sample of free-living adults aged 35 and older [68]. The algorithm, using a 24 h wear protocol, automatically classifies activity events recorded by the activPAL into sleep or non-wear and valid or non-valid days [68]. To be included in the analysis, four days of valid wear with a minimum of one non-work day was required [67]. Sleep and times when the device was not worn were excluded from the analysis. A valid day was considered as ≥10 h of wear time, ≥500 steps, and <95% of time spent in one activity (standing, sitting, walking) [67]. For workdays only, devices had to be worn ≥80% of the time at work and for ≥5 h of working hours in order to be included in the analysis. The outcome variables generated were time spent sitting, standing, and stepping; number and time spent in prolonged sitting bouts (i.e., sitting bouts ≥30 min in duration); number of breaks in sitting; and number of steps. Sitting data are presented as percentage of total work time or percentage of total daily time normalised to an 8 h workday or a 16 h waking day, respectively. Results are presented as mean (M) and 95% confidence interval (95% CI) or standard deviation (SD).

Anthropometric, cardiometabolic risk marker, and psychometric measures

Height was measured to the nearest 0.1 cm (Leicester height measure; Seca, Hamburg, Germany). Waist circumference was measured halfway between the lowest rib and iliac crest to the nearest 0.1 cm (Seca 201; Seca, Hamburg, Germany). Weight, fat mass and fat-free mass were measured to the nearest 0.1 kg, and body fat percentage to the nearest 0.1% using a bioelectrical impedance device (TANITA BC-418MA Segmental Body Composition Analyzer; Tanita Corp., Tokyo, Japan). The average of three blood pressure measures taken after five minutes’ rest was recorded (Omron M5-I; Omron Matsusaka Co Ltd., Matsusaka, Japan). Mean arterial pressure [69] was calculated as:$$\mathrm{MAP}≅{\mathrm{Pressure}}_{\mathrm{Diastolic}}+\frac{1}{3\left({\mathrm{Pressure}}_{\mathrm{Systolic}}-{\mathrm{Pressure}}_{\mathrm{Diastolic}}\right)}$$

Overnight fasted total cholesterol, low-density lipoprotein cholesterol (LDL), high-density lipoprotein cholesterol (HDL), triglycerides and blood glucose were measured via a fingerprick blood sample using a Cholestec-LDX analyzer (Abbott Laboratories, Chicago, IL, USA). Participants were asked to refrain from caffeine and alcohol for 24 h prior to these measures being taken.

Participants completed the following psychological questionnaires: Operational and organisational police stress questionnaires [70]; Warwick–Edinburgh mental well-being scale [71]; positive and negative affect schedule [72], and self-rated job satisfaction and job performance using two single-item questions [73,74]. Letters were sent to participants recommending that they refer themselves to their doctor if they had any cardiometabolic risk marker readings outside of recommended National Institute for Health and Care Excellence (NICE) guidelines.

Post-intervention interviews

Semi-structured interviews were conducted by MLB to explore acceptability and feasibility of the intervention after all follow-up measures were taken. Participants were informed of the purpose of the interview and provided verbal consent to participate. Interview questions were designed around the capability, opportunity, motivation-behaviour model (COM-B) [44,45] and adapted from Ojo et al. [75] (see Appendix A for the interview schedule). All participants who completed the study were interviewed (*n* = 19), but one recording was lost due to technical issues and is therefore not included in the analysis. Interviews were recorded and transcribed verbatim using Otter.ai (AISense, Inc., Los Altos, CA, USA). Otter.ai transcriptions were checked and corrected for accuracy. Transcripts were de-identified and anonymised prior to analysis.

### 2.3. Data Analysis

#### 2.3.1. Primary Outcome—Feasibility and Acceptability

Participation rates are presented as number (*n*) and/or percentage (%). The PRECIS-2 tool and RE-AIM QuEST frameworks were used to narratively evaluate intervention pragmatism and feasibility/acceptability, respectively. PRECIS-2 ratings were narratively assessed first by MLB and then discussed and agreed by all members of the research team (LRS, AMC, DPB).

For RE-AIM, recruitment, retention, and missing data rates are quantitatively presented in absolute numbers (*n*) and percentages (%). To assess acceptability, interview transcripts were imported into NVivo software (version 11, QSR International Pty Ltd., Melbourne, Australia). Transcripts were thematically analysed (by MLB and AMC) using inductive methods to identify key themes and subthemes [76]. Key themes were ‘charted in’ to RE-AIM QuEST using framework analysis [77], checked and agreed by all authors. Illustrative quotations were used to provide context around the RE-AIM evaluation [78].

#### 2.3.2. Secondary Outcome—Potential Effects

For potential effects of the intervention, a paired samples *t*-test was performed to assess changes from baseline to follow-up (with 95% confidence intervals [95% CI]) using SPSS v26.0 (SPSS Inc., Armonk, NY, USA). Missing data were excluded case-wise. Significance was set at a two-tailed alpha level of *p* ≤ 0.05. Cohen’s *d* was calculated to describe the magnitude of change [79], with *d* ≤ 0.2 considered a small effect, *d* = 0.5 a medium effect, and *d* ≥ 0.8 a large effect [80].

## 3. Results

Participation rates and sample description

Figure 3 shows the flow of participants through the study [38].

The cohort was predominantly white British (87%), female (79%), non-managers (67%), with an average of 12 ± 11 years in service (see Table 2). Participants worked an average of 38.2 ± 1.9 h per week on 7.5 h shifts (range: 7.5–12.0 h).

### 3.1. Primary Outcome—Feasibility and Acceptability

#### 3.1.1. PRECIS-2: How Pragmatic Was the A-REST Intervention?

The A-REST intervention was assessed as a largely pragmatic trial according to PRECIS-2 (rated 4 ± 0.9 out of 5) (see Figure 4). The nine domain ratings and their justifications are presented as Domain—Rating (*Guiding question*) followed by the rating justification.

1.Eligibility—Rated 4


*To what extent are the participants in the trial similar to those who would receive this intervention if it was part of usual care?*


This study’s sample characteristics (e.g., age, sex) were comparable to other sedentary workplace interventions [8,9,10]. Intervention participants were largely representative of the type of individual who would receive this treatment if it was usual care, namely, police staff. A higher proportion of women (79.2%, *n* = 19) than men participated in the study, which was a greater representation than in the total staff population at the organisation (~64.0%) [81]. Additionally, 12.5% of participants (*n* = 3) in this study were people from ethnic minority backgrounds, which was a greater representation than in the police staff population (5.3%) [81]. Due to self-selection and the fact the trial was conducted at only a single site, the rating was lowered one point to 4.

2.Recruitment—Rated 3


*How much extra effort is made to recruit participants over and above what would be used in the usual care setting to engage with patients?*


Several recruitment efforts including targeted emails, management assistance, booths in the canteen, and multiple intranet postings were made to recruit participants. This would be over and above that which would be expected in a usual care scenario if it was run internally by the police force. However, emails and intranet postings were not onerous and the Force already employs a Health and Wellbeing coordinator who could support these activities.

3.Setting—Rated 4


*How different are the settings of the trial from the usual care setting?*


The trial was conducted in a range of departments and offices where police staff work. However, it was only conducted in one constabulary site.

4.Organisation—Rated 4


*How different are the resources, provider expertise, and the organisation of care delivery in the intervention arm of the trial from those available in usual care?*


The intervention is likely to be relatively easily transferable into practice with minimal training of providers and by automating certain aspects of the intervention e.g., the sitting breaks competition.

5.Flexibility of delivery—Rated 4


*How different is the flexibility in how the intervention is delivered and the flexibility anticipated in usual care?*


The intervention was flexible and potentially scalable. QR codes can be placed wherever is useful within an organisation and the leaderboard can be automated in future. The intervention can be delivered by an in-house provider with minimal training.

6.Flexibility of adherence—Rated 3


*How different is the flexibility in how participants are monitored and encouraged to adhere to the intervention from the flexibility anticipated in usual care?*


The intervention is not as flexible as it could be because of the person power required to run the competition, compile the leaderboard, and send out emails each week, which would be over and above usual practice. Automation of the leaderboard and weekly emails would improve flexibility.

7.Follow-up—Rated 2


*How different is the intensity of measurement and follow-up of participants in the trial from the typical follow-up in usual care?*


Some of the trial measurements may need adding or adapting to usual care (e.g., activity monitoring, some health outcomes), thus making the trial more explanatory.

8.Primary outcome—Rated 5


*To what extent is the trial’s primary outcome directly relevant to participants?*


The expected primary outcome for a main trial (workplace sitting) was perceived to be highly relevant to individuals in the current study.

9.Primary analysis—Rated 5


*To what extent are all data included in the analysis of the primary outcome?*


The intervention was designed to be feasible in a large population, thus in future trials, intention-to-treat analyses using all available data would be advised.

#### 3.1.2. Reach, Effectiveness, Adoption, Implementation, and Maintenance (RE-AIM)

Reach

Recruitment and retention rates are presented in Figure 1. About one-quarter (*n* = 46) of target participants (*n* = 175–200) expressed interest in the study, with half of these meeting eligibility requirements and enrolling (*n* = 24). Thus, the intervention had a conservative estimate of reach of 12%. The Communications team posted the study on the Force intranet (potentially viewable to all employees) towards the end of recruitment to boost numbers. This strategy appeared to prompt employees from the originally targeted population into registering their interest nearly two months after the recruitment process began.

Effectiveness

This domain of RE-AIM was not relevant for this feasibility study, but potential effects on secondary outcomes are reported below to help inform the efficacy of the intervention that could be evaluated in a full RCT.

Adoption (by people who deliver the programme)

Eleven departmental managers at the organisation were contacted by the research liaison. All gave permission for their employees to be approached to participate in the intervention.

Adoption (by settings)

Adoption by departments was mixed at the single site level. Nine out of eleven (82%) departments initially contacted for recruitment participated, although a third of these were represented by only one employee (see Figure 1—recruitment flow chart). The two departments that did not have any employees volunteer to take part were Information and Communication Technologies and the Resource Management Unit. This was due to these workers working primarily at home and/or away from headquarters the majority of the week (in satellite offices, for example).

Implementation (by people who deliver the programme)

It was originally intended that one participant from each office team would volunteer as a health champion in order to personalise and send the weekly intervention support emails. The small teams of three were not conducive to this approach, thus the research liaison (who was not a participant in the study, but who did sit in one of the participating offices), was asked to perform the role. Due to the time requirement, the liaison was only able to personalise two of the eight emails and only the first of the eight emails was sent to participants by the liaison (low fidelity to protocol). The remaining emails were sent to participants by the research team.

Implementation (individual)
*Education session*To accommodate varying work shift patterns and availability, the education session was offered at four different times over four days during week one of the intervention. Five participants were unable to attend any education sessions and were thus hand-delivered the A-REST booklet and education session materials (a handout with suggestions for electronic prompt tools) by the research liaison. For those who attended, the educational lecture was delivered as intended and appeared useful for providing information about health consequences:
“*I was surprised by the basis for it. How detrimental the sitting, excessive sitting is. I hadn’t really thought about it*”.(P5)*A-REST booklet*The booklet was completed by participants as intended during the education session, but it was not particularly memorable as indicated during the interviews; the education session itself was remembered better. Participants generally found the booklet useful, but most did not use it beyond the education session:
“*It was probably more useful going through it in a big group. I thought that was quite good in that but honestly, I didn’t refer to it afterwards*”.(P1)Participants who did not attend an education session most likely did not complete the booklet. Managers stated that they participated in the study to demonstrate solidarity with employees:
“*To be honest, it [the booklet] wasn’t overly relevant. Just because of what I’ve said earlier I was mostly more focused on the health bit and making sure the organisation are taking part rather than myself*”.(P17)*Electronic prompts*To begin with, implementation of some intervention components had to be adapted due to security permissions. For example, because free computer prompt software options were open source, it was not possible to use these in a high security setting, and there was neither the time nor the resources to procure closed source software. The intervention was adapted by instructing individuals to choose from a selection of electronic prompts available on their phone or computer (e.g., phone app, recurring calendar appointment, or alarm). It was hypothesised that if the participant chose their preferred mode of prompting, they would be more likely to engage with it. Engagement with the prompts varied based on the individual but according to the interviews, the majority of participants used a prompt tool or were prompted to break up their sitting by their colleague’s prompt tool or behaviour:
“*I think having several people in the office did, it really helped because as I said, especially those that are more regimented and sat at their desk, they would have an alarm set. They would get up every, so, literally one person would get up and all of us would go, ‘It’s time!’ and follow each other up*”.(P14)Over time, participants began to anticipate the prompt showing improved awareness of time spent sitting:
“*As time went on, I kind of got used to it and then I was looking for it, even before it pinged, if you know what I mean. But, and I think the QR code really helped in like giving you a reason to get up*”.(P8)*QR codes*Participants indicated that the most frequently used QR codes were those located within their respective offices:
“*Just it [the office QR code] was the closest by. I did use the one in the canteen a few times. And I used the one that I had on the card as well for like the meetings*”.(P9)Evidence of behaviour patterns emerged; for example, a couple of participants regularly walked during their breaks and would scan the same QR codes in the order that they passed them. Participants still found it challenging to take breaks and/or stand during meetings:
“*Yeah, I’ll have a meeting and, well, we said this before about maybe standing up, but yeah in practice it doesn’t work so much*”.(P4)*Competition*The competition helped make the taking of breaks acceptable because of the team-based nature of it:
“*I like that camaraderie that, ‘Let’s do it together. Let’s motivate each other*”.(P15)A complaint of cheating made in a joking manner over email was explored by the research team in the second week:
“*They cheated!*

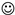
”.(P9)QR code break logs checked by the researcher (MLB) confirmed that two participants were repeatedly logging breaks less than 30 min apart. Once the pattern was detected, the decision was made to cap the number of breaks a participant could claim based on their working hours that day (i.e., 16 breaks in an 8 h workday). The competition also had to be altered due to the very high and very low engagement of certain teams. It was felt that large discrepancies in points between these teams would lead to amotivation among low engagers and therefore the decision was made to average the team points based on participating individuals within the team each week. Interviews revealed that there were occasional issues around internet connectivity and specific devices that affected their ability to log their sitting breaks:
“*Sometimes mine didn’t register. Because I was going to scan it and I thought I couldn’t connect, particularly downstairs. Or sometimes, my phone’s not brilliant, it wasn’t connecting to wherever it should connect, so it’d scan but then it gets stuck*”.(P4)*Emails*Participants reported looking forward to the emails each week in large part due to finding out the leaderboard standings:
“*Yeah I liked that. I think that was good and everybody was like, ‘Oh, [colleague’s name] you’ve won again on that team. ((laughs)) Something like that that’s fine, you know, a little bit of fun isn’t it? And a little trophy. ‘Oh yes, we’ve got the trophy’. Yeah so that was good*”.(P18)No adaptations were made to the intervention apart from the research liaison adding a personalised message in weeks one and five. Participants did not feel the presence or absence of management’s involvement with the emails impacted how they perceived the message, nor did it affect their engagement with the rest of the intervention. It was generally remarked that they felt supported by the organisation:
“*I think it’s always good to see little messages and things. That little extra inspiration or something*”.(P18)*Rise and recharge^®^ app*Most participants reported downloading and engaging with this sitting break tracking smartphone app:
“*I quite like the way it laid out like the colours and the circles and things. Quite visual it was easy to see how you’ve done in the day*”.(P13)Importantly, the app only gave feedback on break behaviour. The optional prompting function was not operational throughout the entirety of the study and thus was not used by participants. An issue that was sometimes reported was accuracy, but it was unclear if this was due to app performance or participants taking breaks without their phone:
“*I did wonder whether it was that accurate because there were some times where I swear I’d got up and it hadn’t registered. And I think in that sense, the biggest problem that I had, ‘cos I don’t really have pockets in my trousers, so I’ll just have my phone on my desk so if I do just nip to the toilet or something, I’d often forget to take the phone with me*”.(P13)*Study measurements*Referral letters were provided after baseline measures to participants with cardiometabolic risk marker readings outside of NICE health guidelines (*n* = 13). A couple of participants remarked on receiving the letters, stating that they impacted their other health behaviours such as nutrition and health seeking behaviour:
“*My cholesterol was better [at the end] which I was a little bit worried about. And so I did make some dietary changes as well. […] probably, after I received my letter about my cholesterol. So then I decided to just make a few changes*”.(P10)”*It’s [my cholesterol’s] gone down. So I did have some blood tests last week. So I did show that letter, that is the email that you sent me, I printed it out about my cholesterol. […] Then they were doing like a full body MOT [health] check*”.(P12)Maintenance (by individuals)

No follow-up was planned after the intervention had ended. In post-intervention interviews, participants discussed their plans for maintenance which included continuing to take frequent breaks from sitting, although sometimes at a lesser frequency, but also attempting/continuing with other healthy behaviours such as making use of sit-stand hot desks, improving their diet, and discussions around forming a walking group:

“*I don’t think I will do half an hour. I would definitely try and do it in within an hour to be able to get up and go do something. Yeah, I think that would work better for me*”.(P12)

“*That’s something I’ve been doing with the up and down-y desks. A lot more is standing up and doing the shift*”.(P1)

New sitting break habits were established in just eight weeks to the point that one person earned a nickname for getting up often to make hot drinks for colleagues:

“*Generally, I’ll make sure, as I say, I’m making coffee every hour and a half, at least. I’ll be up doing that and obviously taking breaks in between then, so yeah, that is definitely more of a habit*”.(P1)

Others were regretful that they had not done more initially to form new habits:

“*I mean I would like to [continue] however, in reality, I probably should have bought into the kind of psychological techniques that you suggested early on… largely because of the kinda the extra strain that I was under*”.(P12)

Maintenance (by the organisation)

Long-term follow-up was not assessed in this feasibility trial; however, feedback following a presentation by AMC and MLB to the police Health and Wellbeing Board in February 2020 indicated that the police force would like to consider continuing with the QR code competition as routine practice.

### 3.2. Secondary Outcomes

#### 3.2.1. Changes in Workplace Sitting, Standing and Stepping

During work time, there were significant differences from baseline to end of intervention (normalised to an 8 h workday). This included decreased sitting time −17.65 min per workday or −3.68% of the workday) and increased standing time (15.49 min) with medium effect sizes (see Table 3). Percentage of time spent in prolonged sitting bouts decreased significantly by 13.32% with a corresponding significant reduction in time spent in prolonged sitting bouts by 63.95 min/work shift; these differences had large effect sizes. The number of prolonged sitting bouts also significantly decreased by 0.96/work shift with a medium-large effect size and the number of sit-upright transitions significantly increased by 3.70 with a medium effect size. Stepping time and number of steps per shift did not differ significantly between baseline and end of intervention.

#### 3.2.2. Changes in Daily Sitting, Standing, and Stepping

There were no significant changes in any daily sitting, standing or stepping variables (normalised to a 16 h waking day) from baseline to end of intervention (see Appendix A).

#### 3.2.3. Changes in Anthropmetric and Cardiometabolic Risk Markers

The data for these variables are presented in Table 4. There were no significant changes from baseline to post-intervention in any anthropometric or cardiometabolic risk outcomes apart from a significant 0.86 kg reduction in body weight (trivial effect size).

#### 3.2.4. Changes in Affect, Wellbeing, Stress, Job Satisfaction and Job Performance

Positive affect significantly improved from baseline to post-interventions with a large effect size (see Table 5). No other questionnaire measures changed significantly.


ijerph-19-09186-t003_Table 3Table 3Changes in sitting, standing, and stepping at work (normalised to an 8 h workday) from baseline to end of intervention (*n* = 15).VariableBaseline Mean (*n* = 15)95% CIPost Intervention Mean (*n* = 15)95% CIMean Difference95% CI
*p*
Effect SizeSitting time (minutes)399.21379.29, 419.14381.56358.78, 404.34−17.65−34.17, −1.130.040.46Sitting time (%)83.1779.02, 87.3279.4974.75, 84.24−3.68−7.12, −0.240.040.46Standing time (minutes)52.8636.84, 68.8968.3648.15, 88.5615.491.87, 29.120.030.47Standing time (%)11.017.67, 14.3514.2410.03, 18.453.230.39, 6.070.030.47Time in sitting bouts ≥ 30 min249.85218.99, 280.71185.89135.72, 236.07−63.95−98.59, −29.31<0.010.85Time in sitting bouts ≥ 30 min (%)52.0545.62, 58.4838.7328.27, 49.18−13.32−20.54, −6.11<0.010.85Number of sitting bouts ≥ 30 min4.704.25, 5.143.732.78, 4.69−0.96−1.80, −0.120.030.72Number of sit-upright transitions22.8119.55, 26.0826.5123.27, 29.763.701.39, 6.02<0.010.63Stepping time (minutes)27.9219.52, 36.3330.0822.45, 37.712.16−6.64, 10.960.610.15Number of steps25921689, 349527111848, 3574119−831, 10690.790.07Abbreviations: CI = confidence interval. Note: The analysis was conducted using 15 complete datasets (activPAL data provided both at baseline and follow-up).



ijerph-19-09186-t004_Table 4Table 4Changes in anthropometric and cardiometabolic risk markers from baseline to post-intervention.Variable (Units)
*n*
Baseline Mean95% CIPost-Intervention Mean95% CIMean Difference95% CI
*p*
Effect SizeWaist circumference (cm)1988.2182.44, 93.9887.7776.12, 99.42−0.44−2.05, 1.170.570.04Weight (kg)1978.4769.98, 86.9577.6157.82, 97.40−0.86−1.68, −0.030.040.05Body Mass Index (kg/m^2^)1927.5925.15, 30.0327.3420.44, 34.25−0.25−0.75, 0.250.320.05Body Fat (%)1934.2429.83, 38.6533.7719.24, 48.30−0.47−2.35, 1.410.600.05Fat Free Mass (kg)1951.1745.34, 57.0150.9729.67, 72.27−0.20−1.53, 1.130.760.02Systolic Blood Pressure (mmHg)19125.42117.39, 133.45124.8189.27, 160.34−0.61−5.70, 4.470.800.04Diastolic Blood Pressure (mmHg)1983.4477.77, 89.1182.9357.95, 107.91−0.51−4.74, 3.720.800.05Resting Heart Rate (bpm)1965.1258.46, 71.7863.4635.41, 91.50−1.67−7.54, 4.210.560.13Total Cholesterol (mmol/L)184.944.43, 5.454.902.95, 6.86−0.03−0.41, 0.340.860.04HDL cholesterol (mmol/L)191.511.21, 1.801.52−0.20, 3.240.01−0.14, 0.160.860.02Triglycerides (mmol/L)191.130.77, 1.491.18−1.13, 3.490.05−0.19, 0.290.650.07LDL cholesterol (mmol/L)133.002.39, 3.622.74−0.14, 5.63−0.26−0.67, 0.160.200.30Non-HDL cholesterol (mmol/L)163.563.03, 4.093.450.27, 6.62−0.11−0.48, 0.250.520.12Glucose (mmol/L)194.954.69, 5.224.752.91, 6.59−0.20−0.45, 0.040.100.41Mean Arterial Pressure (mmHg)1897.4291.22, 103.6196.8942.11, 151.66−0.53−4.77, 3.710.800.04Abbreviations: HDL = high-density lipoprotein cholesterol, LDL = low-density lipoprotein cholesterol, CI = confidence interval.



ijerph-19-09186-t005_Table 5Table 5Changes in affect, wellbeing, occupational and organisational stress, job satisfaction and job performance from baseline to post-intervention (*n* = 19).VariableBaseline Mean95% CIPost-Intervention Mean95% CIMean Difference95% CI
*p*
Effect SizePositive affect28.8425.14, 32.5532.4729.69, 35.253.630.89, 6.370.010.87Negative affect14.3211.66, 16.9714.4711.98, 16.960.16−1.74, 2.060.860.04Wellbeing48.5845.58, 51.5849.8947.31, 52.471.32−1.39, 4.020.320.20Occupational stress34.3726.70, 42.0434.4728.51, 40.430.11−8.11, 8.320.980.01Organisational stress41.5330.25, 52.8144.7432.04, 57.443.21−7.44, 13.860.540.17Job Satisfaction5.054.48, 5.625.164.65, 5.670.11−0.32, 0.530.610.10Job Performance5.475.01, 5.935.635.23, 6.030.16−0.13, 0.450.270.18


## 4. Discussion

### 4.1. Feasibility and Acceptability of the Intervention

The A-REST intervention was feasible to implement and demonstrated good acceptability among police staff. In terms of feasibility, the study was assessed as highly pragmatic, had a 12% reach among the targeted population, was adopted across a variety of departments, and was largely implemented as intended, though there was potential for unintended BCTs and spill-over effects. In terms of acceptability, the intervention had 80% participant retention, produced positive changes in sitting behaviour at work and positive affect, and received favourable and constructive interview feedback.

Pragmatism

Most aspects of the study, according to the PRECIS-2 assessment, were pragmatic as it was deployed under usual conditions at the participating organisation with participants who may be representative of typical police staff [13,82]. This demonstrates that a multi-component intervention with an education session, QR code competition, electronic prompts and weekly emails, was realistic for this population and setting. The higher proportion of women in the present study (than that in the overall study site) may be because women are more likely to self-select for sedentary workplace intervention studies [8,9,10]. Additionally, 12.5% of the participants in this study were people from ethnic minority backgrounds, which was greater than the representation of these individuals in the general police staff population at Bedfordshire [81]. During recruitment, participants were encouraged to sign up with colleagues in the same office because of the team-based component. Thus, in terms of eligibility, the social support aspect of study recruitment and intervention might partly explain the over-representation of these groups. However, given the small sample size of the present study, further research is needed to determine if a teams-based approach is beneficial for inclusive recruitment and retention efforts. Pragmatically, intervention designers could consider using small teams-based interventions to take advantage of the potential to attract colleagues from under-represented groups. Future trials should also consider using stratified sampling and recruitment efforts to reflect the demographic makeup of the workforce.

This is the first multi-component intervention aiming to reduce and break up sitting by implementing a team-based QR code competition. Interventions where entire worksites, offices or floors of a building participate together has been found to encourage social support [83] and beneficially impact workplace social dynamics to facilitate behaviour change [84]. Specific to police, a competitive social environment has been found to promote camaraderie and increase morale in an intervention to reduce sitting and increase physical activity [85]. Qualitative accounts in the present study suggest that the QR code team competition and localised prompts where employees got up together were key drivers for behaviour change. In developing the A-REST intervention, it was identified that a team-based intervention with shared goals around sitting reduction would help address a workplace culture synonymous with sitting [45]. Furthermore, a review of health promotion strategies (targeting a range of behaviours such as physical activity, sedentary behaviour, diet, and others) in the police found that multi-component interventions with a peer support component beneficially impacted more health outcomes (e.g., blood pressure, stress, tobacco use) compared to those without peer support [30]. From a pragmatic standpoint, the competition and other intervention components were flexible and potentially scalable. QR codes can be placed wherever is useful within an organisation and the leaderboard can be automated to make it more pragmatic. The competition can be delivered by an in-house provider with minimal training. It is thus recommended that interventions to reduce and break up sitting in police staff include a competitive, team-based component, and that QR codes be considered as a pragmatic, feasible and acceptable strategy.

Reach and adoption

A reach of 12% in the police setting was comparable to other sedentary workplace interventions [21,84,86,87,88,89]. The targeted recruitment strategy was shown to be feasible and is recommended for future interventions in this target group.

Adoption rates showed that the intervention was feasible across a range of police staff and departments, although uptake was low in some departments. This could be due to eligibility criteria limiting the participation of part-time workers, those with active/mobile duties, and those who work across different office sites during the week. Increasing adoption in police staff may require a more inclusive set of eligibility criteria when scaling up.

Implementation

Qualitative perceptions around engagement with the intervention found that frequent short breaks in sitting were feasible to implement and acceptable to participants. This is important because productivity concerns can be a barrier for police staff in reducing and breaking up sitting [45]. In addition, the sitting breaks competition was well-received by participants and appeared to be feasible for use across different departmental offices. Receiving the emailed leaderboard standings every week was highly motivating for the participants. Electronic prompts were also effective for improving awareness of time spent sitting and for reminding participants to take a break from sitting to the point where individuals were anticipating the prompt. When implementing electronic prompts, participants demonstrated individual preferences for a range of options including phone alarms, calendar reminders, and smartphone apps. These results add to the growing number of sedentary behaviour intervention studies tailoring interventions to individual preferences, while still delivering a core set of BCTs [90,91]. Furthermore, the current study’s findings align with previous evidence supporting the use of self-monitoring tools for reducing sitting [74,92,93]. The use of a QR code competition, individualised electronic prompts, and a self-monitoring phone app were found to be acceptable for supporting behaviour change and these should be considered for scaling up this kind of intervention.

Participants expressed appreciation for the organisational support demonstrated by personalised weekly emails sent out by management. However, management did not realistically have the time to write and send personalised messages every week. A pragmatic solution would be to ask management to brainstorm a range of short, personalised messages at the outset of the intervention (perhaps as part of the education session), which could be transferred onto the email template and scheduled for automated delivery each week. This pragmatic approach should be considered when scaling up interventions that utilise organisational support in this way.

Unintended effects

Data collection was identified as providing unintended BCTs for some participants, such as Monitoring of behaviour by others without feedback, Monitoring without feedback, and Biofeedback. Repeated measurements may influence participants’ motivation [94]. Doctor self-referral letters were provided to half of the study participants, which may have made those individuals more aware and motivated to change their sitting behaviour. Spill-over effects were noted by participants as positively impacting nutritional intake and other health seeking behaviours (unintended BCTs: Information about health consequences, Salience of consequences, Feedback on outcomes of behaviour, Prompts/cues, Incompatible beliefs and Comparison of outcomes). The present study extends knowledge in relation to how data collection procedures may deliver unintended BCTs. Future studies should consider if the delivery of unintended BCTs could differentially affect behavioural outcomes of both control and treatment study arms.

Maintenance

Maintenance was not evaluated in this feasibility trial. However, management feedback to the research team revealed that the QR codes were still present in the organisation two months after the trial and that there were plans to investigate how the organisation could make the competition routine practice. Participants talked about continuing with their new sitting habits and starting up a walking group. Long-term follow-up of individual and organisational maintenance behaviours should be evaluated in a full-scale trial.

### 4.2. Potential Effects of the Intervention

In the present study, sitting time at work significantly reduced from baseline to end of the intervention. Sitting appeared to be replaced predominantly with standing. According to interviews, the most frequently accessed QR codes were those in participants’ offices, thus, they did not have to walk far and may have spent most of their break time standing. Interventions targeting (and reporting on) sitting outcomes are understudied in the police and it is therefore difficult to make comparisons. The PAW-Force trial [85], a 12-week mobile health (mHealth) intervention in British police officers and staff, involved the use of a Fitbit^®^ activity monitor and ‘Bupa Boost’ smartphone app to deliver individual components (e.g., behavioural goal-setting, self-monitoring), as well as social components (e.g., social comparison, competition, social support). However, the intervention did not lead to changes in sitting (even after a further 5 months of optional use) [85]. Other studies where standing has largely replaced sitting have used sit-stand desks [21,27,74,84,95]. Where sit-stand desks are not provided, interventions often incorporate step count competitions leading to sitting being replaced with stepping (or walking) [73,87,96]. In prompt-only interventions (when neither step challenges nor sit-stand desks have been provided), studies have had mixed success in reducing sitting [21,97,98]. A full-scale trial is needed to evaluate the effectiveness of the present study’s multi-component intervention, which did not incorporate sit-stand desks or a step challenge, for reducing and breaking up sitting in police staff.

Despite there being reductions in sitting at work, the lack of change in total daily sitting may suggest a compensation effect outside of work hours whereby participants sat more during non-work hours [99,100]. Future studies should evaluate whether there is a compensation effect in response to similar interventions [100]. It may also be useful to adopt a whole systems approach [101] recognising that sedentary behaviour occurs in three key domains: transportation, occupation, and recreation [102]. Relatedly, a national and global whole systems approach has been recommend for increasing physical activity in the population [103], and a similar principle could be considered for reducing sedentary behaviour. In scaling up this intervention, researchers should consider providing behaviour change support for domains both inside and outside of work to support reductions in sitting across the whole day.

There were limited effects of the intervention on cardiometabolic risk markers, although body weight decreased by 0.86 kg on average. The timeframe of eight weeks in the present study may not have been sufficient to achieve improvements in other cardiometabolic risk markers. Sedentary workplace interventions have shown promise for improving cardiometabolic risk markers, but studies suffer from small samples sizes, low quality, and inconsistencies across risk markers assessed and outcomes achieved [10]. Short-term studies (≤3 months) indicate that cardiometabolic risk marker change is possible, but measures taken are inconsistent [10]. It is difficult to draw conclusions about longer-term studies (≥12 months) as few have been published. Stand up Victoria! found improved cardiometabolic risk scores in the intervention group relative to controls after 12 months, but this was due to control group worsening over time [23]. Thus, sustained sedentary behaviour change may beneficially impact employees by maintaining present health status [23]. Longer-term studies that are adequately powered for detecting cardiometabolic risk marker changes are warranted.

### 4.3. Strengths and Limitations

Limitations of the study involved threats to internal validity. The single-arm, repeated measures design meant there was no control group for comparison of potential behavioural and health effects. A further limitation is that the study was not powered for detecting differences in secondary outcomes. Thus, caution should be exercised when interpreting the results regarding intervention effects. A further limitation was that participants self-selected for the intervention leading to possible bias in the results [104]. Information was not captured on characteristics of those not participating, thus it was not possible to determine the level of selection bias in this study. In a definitive trial, it is recommended to evaluate and address potential selection bias.

Data collection was identified by nearly all participants as providing unintended BCTs, the effects of which may be considered a threat to internal validity. The ‘mere measurement’ effect of repeat testing has been shown to influence sedentary behaviour due to greater awareness of the study purpose [94]. Other limitations of the study involved threats to external validity. The A-REST intervention was conducted at a single worksite among police staff which limits the generalisability of the findings to other occupational subgroups. However, other feasibility and pilot studies have taken a similar approach for evaluating feasibility/acceptability and potential effectiveness [73,84,104,105]. The fact it was conducted under real-world conditions provides high ecological validity around the acceptability and feasibility of the intervention in the police setting [85]. With forty-three police forces in England and Wales employing over 76,000 police staff [106], there is potential to reach a greater number of sedentary workers should the intervention be scaled up [74].

As feasibility and acceptability of the intervention were the primary outcomes of this study, the limitations of this study design were balanced by the greater amount of quantitative and qualitative data provided on the intervention experience [9]. The use of a validated, thigh-worn tri-axial accelerometer and inclinometer device for measuring sitting, standing and stepping was a strength of the study. Self-report measures may be subject to response bias and underestimate sitting time [107,108], and waist-worn, uniaxial accelerometers are limited in their ability to distinguish between different postures [109].

The cohort was predominantly white British (87%) and female (79%), which may limit the generalisability of the findings. Recruitment methods may have an impact on participation levels. The research liaison and the study lead (MLB) were white female, which might have influenced recruitment. The online recruitment methods occurred via emails disseminated by departmental line managers (whose demographic characteristics were not captured) and via e-poster (featuring a white female office worker in focus in the foreground and a white male out of focus in the background) on the organisation’s intranet. Potential participants were encouraged to sign up with officemates/colleagues/friends at work because office team clusters of three people were needed. Also of note, 12.5% of participants in this study were people from ethnic minority backgrounds which was similar to figures (15.7–22%) reported in prior sedentary workplace interventions [23,73,74]. To enhance generalisability, future trials might consider using stratified sampling and recruit using individuals from a range of backgrounds.

The main strength of this study is the comprehensive, mixed-methods assessment of feasibility and acceptability using the standardised PRECIS-2 and RE-AIM QuEST frameworks. The present study also adds to the greater understanding of how and if data collection, as perceived by participants, delivers unintended BCTs. A strength of the study was that all outcome measurements were deemed as feasible and acceptable, though the short-term nature of the intervention remains a limitation. Longer-term studies are needed to evaluate primary and secondary outcomes over time. This is the first study, to the authors’ knowledge, that has used both frameworks simultaneously to comprehensively assess a sedentary workplace intervention trial. The mixed-methods approach also provides a richer understanding of RE-AIM components to better inform the scaling up of A-REST and other interventions in fully powered RCTs.

## 5. Conclusions

The findings from this study suggest that it is feasible to implement and evaluate a theory-based, multi-component sedentary workplace intervention in a police staff setting. The study was highly pragmatic in terms of eligibility, organisation, flexibility for adherence, primary outcome, and primary analysis. Reach and adoption indicators showed that the intervention was feasible with a range of police staff and departments; and had employees who were demographically similar to those in other multi-component interventions. Reach, adoption and implementation were somewhat affected by security procedures at the participating organisation. With regards to implementation, the intervention was delivered mostly as planned with minor deviations from the protocol. There were potential improvements in sitting and standing in response to the intervention. Overall, the findings from this study suggest that a full RCT powered to detect changes in sitting and health outcomes should be conducted to establish the effectiveness of the intervention and its adoption into routine practice in the police setting.

## Figures and Tables

**Figure 3 ijerph-19-09186-f003:**
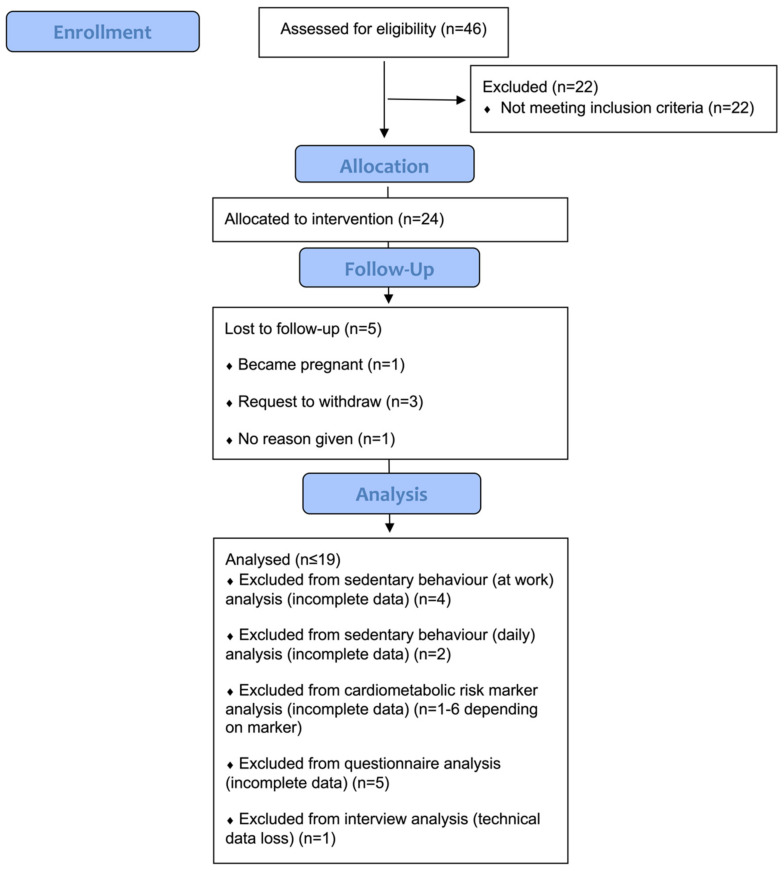
CONSORT diagram showing the flow of participants through the study.

**Figure 4 ijerph-19-09186-f004:**
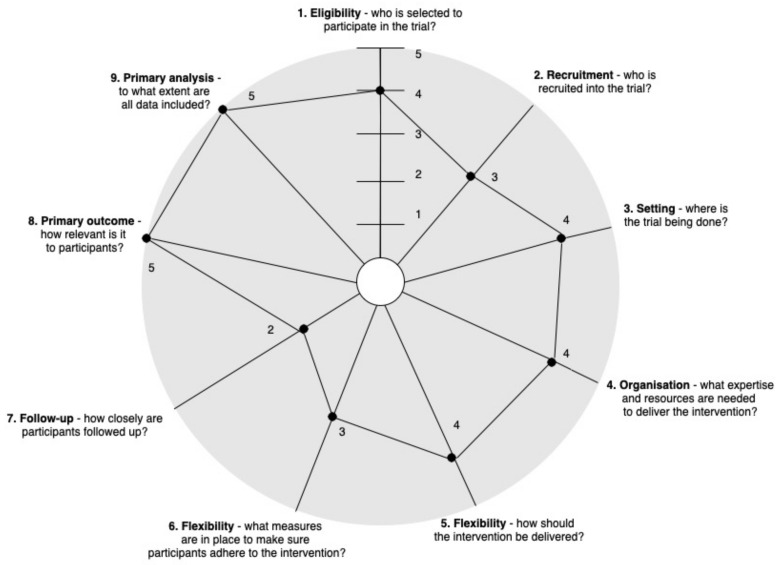
Feasibility evaluation of A-REST intervention according to the PRagmatic Explanatory Continuum Indicator Summary 2 wheel (adapted from [32]).

**Table 1 ijerph-19-09186-t001:** RE-AIM QuEST dimensions with quantitative and qualitative indicators and measures (table adapted from [59]).

RE-AIM Dimension *Guiding Question(s)*	Data Type	Indicator	Measure
Reach *Is the intervention reaching the target population? Those most in need?*	Quantitative	Absolute number, proportion, and representativeness of eligible individuals who participate	Participation rate = #participating/#eligible Overall retention rate = #completed the study/#enrolled Demographic data frequencies and percentages Number (*n*) and percentage (%) of participants providing all and/or partial data
	Qualitative	What explains variation in Reach, number of participants enrolled, and the dropout rate? Questions specific to A-REST: What were the motivations for participating in the intervention?	Semi-structured interviews with participants
Effectiveness *Does the intervention accomplish its goals?*	Quantitative	Intervention effects on outcomes	Potential effects on sedentary behaviour, physical activity, health and wellbeing
	Qualitative	What are the conditions and mechanisms that lead to effectiveness? What are the potential adverse side-effects?	Semi-structured interviews with participants
Adoption *To what extent are those targeted to deliver the intervention participating?*	Quantitative	Number, percentage, and representativeness of participating settings and providers	Uptake = #departments participating/#invited
	Qualitative	What affects provider participation?	Semi-structured interviews with participants
Implementation *To what extent was the intervention consistently implemented?*	Quantitative	The extent to which the intervention was consistently implemented (and delivered as intended)	Device-measured sitting and activity #emails sent
	Qualitative	What were the modifications to the intervention and why did they occur? What are the contextual factors and processes underlying alterations to implementation and how to address them? Questions specific to A-REST: Experience of: education session, team competition, Rise & Recharge^®^ app, electronic prompts, other resources not suggested by the intervention, and weekly emails?	Semi-structured interviews with participants Descriptions of fidelity to protocol (actual versus intended) [58]
Maintenance *To what extent did the intervention become part of routine organisational practices and maintain effectiveness?*	Quantitative	The extent to which a programme becomes part of routine organisational practices/policies and maintains effectiveness	Not assessed quantitatively
	Qualitative	In what form are the components of the intervention or behaviour sustained? Questions specific to A-REST: How, and to what extent, do participants intend to maintain behaviour change?	Semi-structured interviews with participants Worksite board meeting report on sustainability

**Table 2 ijerph-19-09186-t002:** Descriptive characteristics of police staff participants.

Characteristic	All Participants (*n* = 24)
Female, *n* (%)	19 (79)
Age (years), M (SD)	43 (11)
People from ethnic minority backgrounds, *n* (%)	3 (13)
Body mass index (kg/m^2^), M (SD)	27.6 (5.2)
Marriage status, *n* (%)	
Cohabiting	4 (17)
Married or civil status	13 (54)
Single	7 (29)
Education, *n* (%)	
GCSE or equivalent	9 (38)
Vocational qualifications	2 (8)
A levels/Highers or equivalent	7 (29)
Bachelor’s degree or equivalent	5 (21)
Postgraduate qualifications	1 (4)
Job role (manager), *n* (%)	8 (33)
Years in service, M (SD)	11.7 (10.8)
Hours worked per week, M (SD)	38.2 (1.9)
Shift length (hours), M (SD)	8.3 (1.4)
Self-rated heath, *n* (%)	
Fair	5 (21)
Good	13 (54)
Very good	6 (25)
Tobacco use, *n* (%)	
Current smoker	3 (13)
Previous smoker	8 (33)
Smoked daily in the past	7 (29)
Alcohol use score (AUDIT-C), M (SD)	
Women	3.5 (1.7)
Men	6.6 (1.3)
IPAQ weekly METs, M (SD)	1457 (829)
Self-reported sitting time, M (SD)	
Weekdays (hours)	15.5 (6.8)
Weekend (hours)	12.8 (8)
Office size, *n* (%)	
Cell office (one person per room)	2 (8)
Shared room (2–3 people per room)	2 (8)
Small landscape (4–9 people per room)	3 (13)
Medium-size landscape (10–24 people per room)	7 (29)
Large-size landscape (24+ people per room)	10 (42)

Abbreviations: AUDIT-C = Alcohol Use Disorders Identification Test—Consumption; M = mean; METs = Metabolic Equivalents of Task, SD = standard deviation.

## Data Availability

The data presented in this study are available on request from the corresponding author. The data are not publicly available due to restrictions, e.g., privacy of qualitative datasets.

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
