# Peer review of "A-REST (Activity to Reduce Excessive Sitting Time): A Feasibility Trial to Reduce Prolonged Sitting in Police Staff"

_ijerph, 2022, doi:10.3390/ijerph19159186_

Round 1

Reviewer 1 Report

Thank you for an interesting article which I enjoyed reading. There are a couple of suggestions I would like to make:

1. Intro section line 37, please consider giving the range of sedentary time or percentage instead of saying a large proportion.

2. You are talking about multicomponent approaches in line 57 and state that they have greater behaviour effects. In comparison with what approaches are these greater effects?

3. A very important aspect that needs to be considered is Systems Approaches which is not mentioned in the background. It is likely the very reason many interventions fail; we address behaviour in work only or in schools and participants compensate at home. It is worth looking at Rutter at al., 2019 : Systems approaches to national and global physical activity plans. Although the focus here is PA, the systems approach principle remains. There are other studies talking about the importance of SA in behaviour change.

4. Section 2.2.1 line 141 and section 2.2.1 line 210. There is an issue with numbering please check.

It would be clearer if the section 2.2.1 line 141 would provide a clear and simple overview of the intervention which you developed. For example A-REST intervention consisting of education, scheduled sedentary breaks++...etc. otherwise the reader needs to piece together what the intervention consists of exactly. A figure would be great if possible.

5. Line 200, please ref Qualtrics.

6. Line 327 it is indeed unusual that 79% or participants are female when they are a minority in the police force. Might you attempt to understand why the intervention was more appealing to females or why the male participants were not so interested. Were the recruiters male or female? Worth considering.

7. Sections 4.2.1 and 4.2.2 could be clearer. Please remind the reader what working hours are and what the whole day is in this study.

8. Other studies have similar results when attempting to reduce sedentary time or increase PA by having an intervention in one location. What could be done to support participants to translate sitting time changes from work behaviour to outside of work environments? This should to be considered in discussion and plans for the next study.

Author Response

Response to Reviewer 1’s comments

We would like to thank Reviewer 1 for taking the time to review our manuscript and suggest constructive comments.

  1. The specific percentage of sedentary time is now provided. [Line 37]
  2. We have amended the text to read: Multi-level approaches are likely to demonstrate greater behaviour effects than single level interventions (e.g., individual-level or organisation-level only) [28] [Line 76]
  3. In the Discussion section, we have now considered how a systems approach to reducing sedentary behaviour might be useful to consider when scaling up the A-REST intervention. We now explore how workplace interventions may not fully address behaviour in other domains (e.g., commuting, leisure time), with participants potentially compensating for reduced sitting at work by increased sitting outside work. We advise that the national and global whole systems approach, which has been recommend for increasing physical activity in the population (Rutter at al., 2019), should be considered for reducing sedentary behaviour. Thus, in scaling up, interventionists should consider providing behaviour change support for domains outside of work to support concomitant non-work reductions in prolonged sedentary behaviour. [Line 1002]
  4. Amendments include: 1. Section numbering has now been adjusted. [All of Section 2 starting at Line 138]. 2. A sentence describing the intervention in brief has now been provided along with a figure for illustration purposes. [Line 181 & Figure 2]
  5. We have left this in as ‘Qualtrics’ [see Line 251] as a full reference has been provided earlier [see Line 151].
  6. We have amended the Limitations section about how recruitment methods may affect uptake among particular demographic groups and limit the generalisability of the findings. [Line 1370]
  7. To reduce confusion around behaviour changes observed at work and over the whole day:
    1. At work - Section 4.2.1 Changes in work sitting, standing and stepping  now reads: “During work time, there were significant differences from baseline to end of intervention (normalised to an 8-hour workday). This included decreased sitting time (-17.65 minutes per workday or -3.68% of the workday) and increased standing time (15.49 minutes) with medium effect sizes (see Table 3).” [Line 746]
    2. Over the whole day - Section 4.2.2 Changes in daily sitting, standing, and stepping now reads: “There were no significant changes in any daily sitting, standing or stepping variables (normalised to a 16-hour waking day) from baseline to end of intervention (see Additional file, 1: Supplement 8).” [Line 780]
  8. Please see response to Number 3.

Reviewer 2 Report

Thank you for the opportunity to review this intervention study. This study aimed to evaluate the acceptability and feasibility of an 8-week, theory-derived sedentary workplace intervention (single-arm, pre-post design) for police office staff. Overall, the intervention was well designed and executed. I have several comments/suggestions for the authors to consider.

  1. This study aimed to evaluate the acceptability and feasibility of the A-REST intervention. However, the authors did not clearly define acceptability and did not describe how to measure it. The PRECIS-2 framework addresses the intervention's practicality/feasibility (how doable) but not acceptability (whether participants perceived the innovation as satisfactory). Implement Sci. 2017; 12: 108 provides detailed descriptions of these two concepts.  
  2. RE-AIM is an evaluation framework for measuring the public health impact of interventions. Please justify its appropriateness for pilot/feasibility studies. 
  3. Please justify which aspects of RE-AIM QuEST address the concept of acceptability.
  4. 4.2.1 "There were significant differences from baseline to end of intervention, including decreased total sitting time by 17.65 minutes, decreased percentage of workday spent sitting 618 by 3.68%, and increased standing time by 15.49 minutes with medium effect sizes (see 619 Table 3)". Do you mean 17.65 minutes in total over the eight weeks or 17.65 minutes per day?
  5. Data collection is a testing effect. Participants showed improvement in the outcomes at the end of the study because of repeated measurement and increased awareness of the study's purpose. The authors may consider the unintended effects as threats to internal validity. 
  6. Study strengths and limitations. A study needs to work on the representativeness of its study population to increase generalizability. A control group cannot provide greater external validity, but it helps to eliminate threats to internal validity.  
  7. The discussion section could be more concise and focus on the main findings.
  8. The strengths and limitations section needs more work to articulate different ideas.   

Author Response

Response to Reviewer 2’s comments
Thank you to Reviewer 2 for their time in reviewing our manuscript and for their suggestion of constructive revisions.

  1. The Methods section now sets out definitions of feasibility and acceptability (Weiner et al., 2017; Proctor et al., 2011) [Line 275]. We have now stated how data assigned to the indicator ‘Implementation’ from the RE-AIM QuEST framework was used in assessing acceptability. [Line 294]
  2. We have now provided justification for the use of RE-AIM in evaluating pilot/feasibility /small-scale studies for scaling up purposes and wider implementation. [Line 298]
  3. We have now provided justification employing the Implementation indicator from RE-AIM QuEST to address the concept of acceptability [Line 294] including the use of interview data [Line 404]
  4. Now clarified to -17.65 minutes of sitting per workday. [Line 747]
  5. A statement is now included in the Limitations section about the potential threat to internal validity caused by the ‘Mere Measurement’ effect (Voigt et al., 2018, BMC Sports Science, 10:1). [Line 1037]
  6. We have now clarified the Strengths and Limitations section about how threats to internal and external validity might be mitigated. Specifically, how a control group would mitigate threats to internal validity (see Response to #5), and improved representativeness of the sample would address external validity (ie, generalisability) [Line 1369]. Amendments have been made to better articulate these ideas [Section 5.3 - starting at Line 1028]
  7. We have re-focussed the Discussion section to highlight the main findings of the study. [Line 826]
  8. Please see response to Number 6.

Additional proofreading corrections:

  • As per current best practice, the term ‘Black and minority ethnic’ (BME) has now been updated throughout to ‘people from ethnic minority backgrounds’ (‘Writing about ethnicity’, GOV.UK, https://www.ethnicity-facts-figures.service.gov.uk/style-guide/writing-about-ethnicity [Accessed 11 July 2022]).
  • Additional file 1 has been updated to now contain 8 supplements (instead of 7) with the addition of a table showing the behaviour change techniques used in A-REST (now Supplement 3).
  • The CONSORT checklist (Additional file 1: Supplement 1) has been updated.
  • NICE acronym updated [Line 381]